# Improving Automatic Melanoma Diagnosis Using Deep Learning-Based Segmentation of Irregular Networks

**DOI:** 10.3390/cancers15041259

**Published:** 2023-02-16

**Authors:** Anand K. Nambisan, Akanksha Maurya, Norsang Lama, Thanh Phan, Gehana Patel, Keith Miller, Binita Lama, Jason Hagerty, Ronald Stanley, William V. Stoecker

**Affiliations:** 1Electrical and Computer Engineering Department, Missouri University of Science and Technology, Rolla, MO 65409, USA; 2Department of Biological Sciences, College of Arts, Sciences, and Education, Missouri University of Science and Technology, Rolla, MO 65409, USA; 3College of Health Sciences, University of Missouri—Columbia, Columbia, MO 65211, USA; 4S&A Technologies, 10101 Stoltz Drive, Rolla, MO 65401, USA

**Keywords:** deep learning, machine learning, fusion, cascade generalization, melanoma, atypical network, branch streaks, angulated lines

## Abstract

**Simple Summary:**

Supervised deep learning techniques can now automatically process whole dermoscopic images and obtain a diagnostic accuracy for melanoma that exceeds that of specialists. These automatic diagnosis systems are now appearing in clinics. However, the computational techniques used cannot be easily interpreted by the experts using the systems, and they still fail to detect a minority of melanomas. We describe an approach that detects critical irregularities in pigment patterns, a clinical feature, and uses this knowledge to improve deep learning diagnostic accuracy. In this research, we trained a deep learning network to identify visible patterns associated with melanoma. We combine these patterns with a supervised whole-image deep learning method to improve diagnostic accuracy and provide a publicly available dataset with the clinical structures annotated.

**Abstract:**

Deep learning has achieved significant success in malignant melanoma diagnosis. These diagnostic models are undergoing a transition into clinical use. However, with melanoma diagnostic accuracy in the range of ninety percent, a significant minority of melanomas are missed by deep learning. Many of the melanomas missed have irregular pigment networks visible using dermoscopy. This research presents an annotated irregular network database and develops a classification pipeline that fuses deep learning image-level results with conventional hand-crafted features from irregular pigment networks. We identified and annotated 487 unique dermoscopic melanoma lesions from images in the ISIC 2019 dermoscopic dataset to create a ground-truth irregular pigment network dataset. We trained multiple transfer learned segmentation models to detect irregular networks in this training set. A separate, mutually exclusive subset of the International Skin Imaging Collaboration (ISIC) 2019 dataset with 500 melanomas and 500 benign lesions was used for training and testing deep learning models for the binary classification of melanoma versus benign. The best segmentation model, U-Net++, generated irregular network masks on the 1000-image dataset. Other classical color, texture, and shape features were calculated for the irregular network areas. We achieved an increase in the recall of melanoma versus benign of 11% and in accuracy of 2% over DL-only models using conventional classifiers in a sequential pipeline based on the cascade generalization framework, with the highest increase in recall accompanying the use of the random forest algorithm. The proposed approach facilitates leveraging the strengths of both deep learning and conventional image processing techniques to improve the accuracy of melanoma diagnosis. Further research combining deep learning with conventional image processing on automatically detected dermoscopic features is warranted.

## 1. Introduction

The number of cases of invasive malignant melanoma estimated for 2023 in the US was 97,610, a slight decline from the 99,780 cases estimated for 2022, but accompanied by a rise in fatalities from 7650 in 2022 to 7990 estimated deaths in 2023 [1,2]. The projected number of cases of early in situ melanoma for 2023 was 89,070. Melanomas detected and treated at this stage are entirely curable. Since all but a few melanomas are visible in the skin, the case for early detection and treatment is particularly compelling for melanoma.

The number of estimated cases of melanoma is only expected to grow. Projections indicate that melanoma will become the second most prevalent form of cancer by 2040 [3]. This highlights the importance of raising melanoma awareness, implementing prevention measures, and improving the early diagnosis of skin lesions. 

Machine vision techniques incorporating deep learning (DL) used as diagnostic assistants via computer-aided diagnosis (CAD) can contribute to early melanoma detection, and they have shown a diagnostic accuracy equal to or exceeding that of dermatologists [4,5]. DL accuracy depends critically upon large numbers of training examples. 

To aid biomedical image processing, experts have been directly involved in the curation of datasets, as well as directing machine learning researchers to incorporate relevant and useful features into the datasets. One example of such work is the HAM10000 dataset of dermoscopic images [6], which has contributed to the International Skin Imaging Collaboration (ISIC) challenge datasets. These have been released in several iterations since 2016 and include a range of skin lesion-related tasks [7,8,9]. Other datasets have also been used in CAD research, including PH2 [10], which has 200 dermoscopic images with three diagnosis classes, and the Interactive Atlas of Dermoscopy [11], which has over 1000 clinical cases with clinical and dermoscopy images, annotations of specific features, histopathology results, and difficulty levels. 

There have been many surveys that have highlighted both machine learning and deep learning approaches to skin lesion research. An analysis of recent DL and machine learning models for lesion classification trained on public datasets, focused on Artificial Neural Networks (ANNs), Convolutional Neural Networks (CNNs), Kohonen Self-Organizing Neural Network (KNNs), and Radial Basis function-based Neural Networks (RBFNs), is presented in [12]. The survey in [13] presents DL and a combination of DL and conventional models, which also incorporate whole-image hand-crafted feature extraction. The survey concludes that there is a lot of potential in using models to aid diagnostic decisions rather than attempting to replace physicians. Another recent survey [14] gives an overview of different DL methods used for lesion classification with clinical, dermoscopy, and histopathology datasets, highlighting the novelty and limitations of each approach. The survey also discusses the poor generalization ability of DL models across different domains compared to dermatologists. In [15], a mobile application is developed that uses models trained on the HAM10000 dataset for early detection and classification. They also use metadata such as location information to find the UV radiation degree and type of skin in their application. All of the recent surveys and applications, including the ones presented here, focus entirely on whole-image classification and do not rely on the dermoscopically relevant features used by dermatologists in the clinic. This is a gap in the current research, the filling of which is vital for building explainable and interpretable methods and needs to be developed further. 

The segmentation of dermoscopic features has been identified as a particularly challenging task in skin lesion analysis. The only publicly available dataset that has provided masks for five dermatological features is the lesion attribute detection challenge dataset for the ISIC 2018 [6,16] challenge. The dataset used coarse Superpixel-based masks for the features that included the general location of the feature but did not delineate the features precisely. In the follow-up review of the ISIC 2018 challenge [16], it was noted that the dermoscopic attributes were hard to segment and the overall performance was very poor on this task, with the highest Jaccard score being only 0.473. They pointed out that one possibility for aiding in the development of detection methods that use dermoscopic features would be to work on annotating patterns that strongly correlate with the diagnosis of interest. This is what we set out to achieve with our globule segmentation dataset [17] and in the current study. Some research has focused on the automatic identification of clinical dermoscopic structures. These include granularity [18] and blotches [19]. These approaches have used conventional image processing, rather than deep learning, to determine visible clinical features. Conventional imaging processing uses morphology, color statistics, texture measures, and localization techniques to identify these structures and distinguish them from mimics. The time-consuming and intensive work needed to develop databases of a sufficient size for the accurate detection of these objects has hindered progress. 

Therefore, recent successful DL approaches to diagnosing melanoma in dermoscopic images have primarily used whole-image processing [4,5], avoiding the time-consuming annotation and detection of individual melanoma features. However, the prominence of pigment network irregularities in the very earliest examples of melanoma prompted us to analyze these network irregularities.

### Irregular Network

Irregular network, as used here, is a general term that encompasses several types of network irregularities that can be present in melanomas. The most common type of irregular network is the atypical pigment network, defined in a consensus conference as a “black, brown, or gray network with irregular holes or thick lines” [20]. A more recent terminology standardization conference defined the atypical network as a “network with increased variability in the color, thickness, and spacing of the lines of the network that is asymmetrically distributed” [21]. These areas have also been described as brown to black in color, chaotic, and sometimes smudged or out of focus [22]. 

Another irregular network structure in melanocytic lesions that correlates with melanoma is irregular streaks: dark, widened, irregular lines [23]. These are a subset of branched streaks: broadened network lines with incomplete connections. Branched streaks may be found in both benign and malignant melanocytic lesions [21].

Angulated lines constitute a third irregular configuration on a larger scale [24]. Angulated lines are very broad and approximately linear structures that may appear as polygons. These structures often show hazy gray areas or dots within the angulated lines. Examples of these irregular network structures are shown in Figure 1. 

Several studies have found that an irregular network is associated with an increased likelihood of melanoma [20,22,25]. A recent meta-analysis using 182 articles [22] found that the odds ratio (OR) for melanoma in lesions with an irregular network was 1.8 to 9.0. Another study on 200 melanoma cases found that the prevalence of the feature was 55.5% [25] and that it was primarily found in superficial spreading melanoma. 

Before the advent of deep learning models for computer vision tasks, researchers developed specialized feature detection/extraction algorithms to identify relevant dermoscopic features, which would then be leveraged for lesion diagnosis. One example of the early detection of irregular/atypical networks in machine learning extracted textural features to detect atypical network regions in dermoscopy images (annotated by a dermatologist) and was subsequently used for benign versus malignant classification [26]. These techniques have not been further used for melanoma detection in large datasets. 

In a preliminary study, we tabulated irregular networks in lesions missed by a deep learning classifier (probability of melanoma < 0.6) using ChimeraNet [27] to diagnose a separate dataset. We sought to determine the most critical feature that could help correct deep learning errors. Sixty-eight melanomas had a calculated DL probability of less than 0.6 using ChimeraNet [27]. Of these lesions, 46 (68%) had an irregular network, as defined above. This percentage was somewhat above that found in other studies, such as 55.5% in superficial spreading melanomas [25], a difference that could be due to different datasets. However, the finding that perhaps one-third of the lesions that deep learning misdiagnosed had an irregular network provided additional support for the importance of irregular networks in dermoscopic diagnosis. This was surprising to us because it was not rare varieties of melanoma that the DL network missed, but rather lesions with an irregular network that DL did not detect.

In summary, irregular networks are a critical dermoscopic feature indicating melanoma. Because of the importance of this feature in indicating the need for further evaluation, including a skin biopsy, we undertook the development of an irregular network database. Additionally, we developed preliminary benchmarks on the segmentation task using a range of different segmentation architectures. We also introduce a novel method of leveraging DL classification probabilities and mask-based hand-crafted features while using conventional classifiers to improve the final test set accuracy for benign versus melanoma classification. The remaining sections of the article are (2) Methods; (3) Results; (4) Hardware and Software; (5) Discussion; and (6) Conclusion.

## 2. Methods

### 2.1. Datasets

The dataset used in this study was curated using a subset of images from the ISIC 2019 dataset [6,8,9]. Using the metadata provided with the image dataset, we associated the ISIC19 training ground-truth CSV file with the metadata file using an image identifier as the primary key. This primary key was labeled as the ‘image’ field in the provided metadata file. The dataset contains 25,331 images compiled from at least three different pre-existing datasets, including the HAM10000 [6], the BCN20000 [9], and the MSK [8] datasets. Due to the data collection and curation method, many of these images were duplicates of a lesion that were either captured at a different angle or at different times [9]. Some of these images have an additional field called ‘lesion_ID’, which assigns the same image identifier to the same lesion. In the dataset, there are 2084 lesions that do not have an assigned ID, to these we assign the ISIC19 image ID as the lesion ID after removing the ‘_downsampled’ suffix that exists on some images. We treated these 2048 images as unique images in our analysis.

Two separate datasets were created for the irregular-network segmentation and benign versus melanoma classification tasks. Additionally, we use the ‘lesion_ID’ to ensure that our splits for segmentation and classification do not overlap and that images with the same ‘lesion_ID’ do not appear across our training, validation, and test set. 

The segmentation dataset is used to train segmentation models to generate irregular-network masks, and then hand-crafted features extracted from these masks are used in the subsequent classification task. 

#### 2.1.1. Segmentation Dataset

A group of trained undergraduate student annotators first create an initial irregular network mask for each image in the separated segmentation dataset. This set consists of 487 melanoma images (449 unique ‘lesion_IDs’) which have a total of 498 masks. The annotators sought to identify those irregular networks consistent with found in melanoma, but not all possible irregular network structures in the images, some of which are associated with atypical but benign lesions. These initial masks were then verified by a practicing dermatologist [WVS] and corrected if necessary. The annotations follow the definitions provided by the virtual consensus meeting for dermoscopy [20]. 

#### 2.1.2. Classification Dataset

To create the dataset for classification, we first discard all the images in the segmentation dataset from the ISIC 2019 dataset. We then randomly selected 500 melanoma (MEL) images and 100 images per class from the Actinic keratosis (AK), Melanocytic nevus (NV), Benign keratosis (BKL), Dermatofibroma (DF), and Vascular lesion (VASC) classes in ISIC 2019. The dataset has 374 MEL lesion identifiers (IDs), 100 NV lesion IDs, 98 BKL lesion IDs, 89 AK lesion IDs, 76 VASC lesion IDs, and 69 DF lesion IDs. Our classification task has a 1000-image dataset with 806 unique lesion IDs. To train and test our DL classification models, we split this dataset into training, testing, and validation sets. 

Next, morphological and color features of irregular networks are used to classify the skin lesions as either benign or melanoma. These features extend the features developed for vessels proposed by Cheng et al. [28], with additional features modified for irregular network masks. Morphological features, including count, length, width, area, and eccentricity of objects, are computed from the final detected irregular network mask. Color features include the mean and standard deviation of each set of RGB pixels that make up an irregular network object, inside and outside the lesion area. All color statistics are computed in the LAB color space. The lesion mask was generated using the ChimeraNet model by Lama et al. [27].

### 2.2. Models

In our study, we developed a procedure that leverages the advantages of established DL architectures and fusing them with conventional learning models to build a binary classifier for melanoma. We start our pipeline by first building an irregular network segmentation model and used four established architectures: U-Net [29], U-Net++ [30], MA-Net [31], and PA-Net [32], commonly used for segmentation. All the architectures are based on an encoder-decoder structure like the ones in auto-encoders and popularized for medical image segmentation by U-Net. 

The U-Net++ architecture is a modified version of the U-Net architecture that incorporates nested dense convolutions to connect each encoder level to its corresponding decoder level. In addition, the architecture features multiple segmentation branches that originate from different levels of the encoder network. Two variants of the U-Net++ architecture exist: the fast mode and the accurate mode. The fast mode selects the final decoder segmentation output as the final mask, while the accurate mode averages all the segmentation masks to generate the final mask. In our work, we use the fast mode. 

The Multi-scale Attention Net (MA-Net) is a segmentation model developed primarily for liver and tumor segmentation. It introduces two new blocks incorporating a self-attention mechanism, namely, the Position-wise Attention Block (PAB) and the Multi-scale Fusion Attention Block (MFAB). These blocks are designed to capture attention feature maps at both the spatial and channel levels. Specifically, the PAB is intended to obtain the spatial dependencies between pixels in a global view, whereas the MFAB captures the channel dependencies between any feature maps by fusing high and low-level semantic features.

The Pyramid Attention Network (PA-Net) is a semantic segmentation model that utilizes global contextual information. It proposes combining an attention mechanism and a spatial pyramid to extract precise dense features for pixel masking. The combination is achieved by introducing a Feature Pyramid Attention (FPA) module, which performs spatial pyramid attention on the high-level output and incorporates global pooling to enhance feature representation. A Global Attention Upsample (GAU) module is employed to guide low-level feature selection using a global context.

We replaced the encoder stage of each of those architectures with four variants of the EfficientNet [33], specifically EfficientNet-B2, EfficientNet-B3, EfficientNet-B4, and EfficientNet-B5 as the encoder networks. Each model in the EfficientNet series of models is a compound-scaled model of previous model iterations. Therefore, this lets us assess the behavior of each architecture under scaling of the encoder. The decoder is constructed symmetrically based on the encoder network used. All segmentation models used were constructed using the implementation in [34] for Pytorch [35] and were pre-trained on the ImageNet dataset [36]. 

For the next stage of our pipeline, classification, we used three different architectures, EfficientNet-B0, EfficientNet-B1, and ResNet50, which were also pre-trained on the ImageNet dataset [36]. We finally use conventional classifier models such as linear support vector machines (SVM), radial basis function (RBF) SVM, random forests (RF), decision trees (DT), and neural networks (NN) to further enhance melanoma classification in the next stage of our pipeline. 

This method of using classifiers sequentially is a variation of the cascade generalization framework proposed in [37] and the more generalized idea of stacked generalization proposed in [38]. We use the nomenclature from [37] and [39] to describe each “level” of our classification process. In the cascade generalization framework, the predicted probabilities of an initial classifier (level-0 models) are concatenated with the inputs to the classifier and fed into a second classifier (level-1 models). In contrast, stacked generalization involves multiple models (all of which are level-0 models) that are used to generate predictions on the dataset and then concatenated and given to a second classifier (level-1 models) for final classification. A similar deep learning approach termed multi-task network cascade was also proposed in [40] for end-to-end instance segmentation. 

In our method, DL models used for classification constitute the level-0 models, while the subsequent conventional classifiers trained using hand-crafted features concatenated with the initial DL output probabilities are the level-1 models. This approach allows us to improve the performance of melanoma classification by leveraging the strengths of both deep learning and conventional classifiers. An overview of the whole process is presented in Figure 2.

### 2.3. Evaluation Metrics

In our study, we used overall pixel-based metrics on the test dataset to evaluate the quality of our segmentations. These metrics were calculated after thresholding the predicted masks using a threshold of 0.5 and accumulating true-positive (TP), false-positive (FP), true-negative (TN), and false-negative (FN) pixels across the entire test dataset. TP, FP, TN, and FN are calculated as follows,
(1)$$\mathrm{TP}=\Sigma_{i=1}^{I}\Sigma_{j=1}^{N_{i}}𝟙\left(\left(p\left(i,j\right)=1\right)\wedge \left(g\left(i,j\right)=1\right)\right),$$
(2)$$\mathrm{FP}=\Sigma_{i=1}^{I}\Sigma_{j=1}^{N_{i}}𝟙\left(\left(p\left(i,j\right)=1\right)\wedge \left(g\left(i,j\right)=0\right)\right),$$
(3)$$\mathrm{TN}=\Sigma_{i=1}^{I}\Sigma_{j=1}^{N_{i}}𝟙\left(\left(p\left(i,j\right)=0\right)\wedge \left(g\left(i,j\right)=0\right)\right),\;\mathrm{and}$$
(4)$$\mathrm{FN}=\Sigma_{i=1}^{I}\Sigma_{j=1}^{N_{i}}𝟙\left(\left(p\left(i,j\right)=1\right)\wedge \left(g\left(i,j\right)=0\right)\right).$$

In the equations above I is the total number of images in the segmentation test dataset, $N_{i}$ is the number of pixels in the $i^{th}$ image in the test dataset, p(i,j) represents the $j^{th}\;$pixel in the predicted mask for the $i^{th}\;$image, t(i,j) represents the $j^{th}\;$pixel in the ground-truth mask of the $i^{th}\;$image, and p(i,j), t(i,j)∈{0,1}. t(i,j)=1, means that the $j^{th}\;$pixel in the ground-truth mask of the $i^{th}\;$image has been annotated as an irregular network pixel and is not an irregular network pixel when t(i,j)=0. p(i,j)=1, means that the $j^{th}\;$pixel in the predicted mask of the $i^{th}\;$image has been predicted as an irregular network pixel and not an irregular network pixel when t(i,j)=0. 𝟙 is the indicator function defined below,
(5)$$𝟙\left(statement\right)=\{\begin{array}{c} 1,\;if\;statement\;true\\ 0,\;otherwise \end{array},$$
and ∧ is the operator for logical conjunction (and). 

We then use the standard definitions for precision, recall, F1-score, and specificity to calculate these metrics as described in the equations below. We also calculated the per-pixel accuracy over the entire test dataset.
Precision = TP/(TP + FP) (6)
Recall = Sensitivity = TPR = TP/(TP + FN)(7)
F1-Score = (2 × Precision × Recall)/(Precision + Recall)(8)
Specificity = TN/(TN + FP)(9)
IoU = TP/(TP + FN + FP)(10)
Accuracy = (TP + TN)/(TP + TN + FP + FN)(11)

To further evaluate the quality of the masks produced by our algorithm, we used the Jaccard index, also known as the Intersection-over-Union (IoU) metric. The Jaccard index is defined as the ratio of the area of the intersection of the predicted binary mask and the ground-truth binary mask to the area of their union, its definition in terms of TP, FP, and FN can be found in Equation (10). It is a widely used metric in the field of image segmentation, as it provides a measure of the similarity between the predicted and ground-truth masks. 

To assess our deep learning and conventional classifiers for the binary benign versus melanoma classification task, we used the metrics, precision, recall, f1-score, and accuracy. In addition to this, we calculated the area under the Receiver Operating Characteristic (ROC) curve (AUC). We also calculate the AUC when the false-positive rate (FPR) is greater than 0.40, as dermatologists prefer higher sensitivity (true-positive rate) in this region of the ROC curve. This means that it is better for a model to be conservative in classifying a lesion as benign compared to classifying it as melanoma, resulting in fewer missed melanomas. 

Our study used the feature permutation importance method to assess feature importance [41]. This method involves randomly shuffling the input feature order and measuring the decrease in model performance (accuracy). By assessing the decrease in performance after shuffling each feature, we can determine the relative importance of each feature in the model’s predictions. This feature permutation importance score is a model-agnostic score, meaning it can be applied to any machine learning model. However, it is essential to note that this method only provides a relative measure of feature importance and does not consider the possible interactions between features, especially if correlated features exist. 

### 2.4. Training

#### 2.4.1. Segmentation 

For segmentation, the segmentation dataset was first split into a 50% training set, a 20% validation set, and a 30% hold-out test set before overlap handling. After ensuring that the same lesions do not appear across the sets, the splits are, 53% in the training set, 15% in the validation set, and 32% in the hold-out test set. We train the models for 100 epochs with an early stopping patience of 20 epochs based on the Jaccard score over the validation set. During training we used an initial learning rate of $\lambda_{s}=$0.001 for the first 50 epochs and reduced it to $\lambda_{s}=0.0001$ for the last 50 epochs. All models take an input of size 448 × 448 which was randomly cropped from the image. Batch size varied depending on the size of the encoder model used due to hardware limitations on the amount of GPU RAM available for training. Batch size of 6 was used for EfficientNet-B2, a batch size of 4 for EfficientNet-B3 and EfficientNet-B4, and a batch size of 2 for EfficientNet-B5. We used various data augmentation techniques to increase the robustness of our segmentation model to variations in lesion color, shape, and viewing angle. Spatial augmentation techniques included random shifting, scaling, horizontal flips, and vertical flips applied to the randomly cropped input images. Color-based augmentation techniques included random hue and saturation adjustments, contrast-limited adaptive histogram equalization (CLAHE), random brightness adjustments, random gamma adjustments, and equalization applied to the input images. 

Other image augmentation methods, such as the addition of random noise, pixel dropout, image sharpening, and blurring were also applied to the input images. Additionally, we randomly performed perspective and elastic transformations to add variations in viewing angles and to account for the elastic nature of human skin. These techniques were used to artificially increase the training sets diversity and improve the model’s generalization capabilities.

The loss used for training the segmentation model is the Dice Loss [42], which has shown great promise in the domain of medical image segmentation [43]. This loss function is particularly well-suited for image segmentation tasks as it measures the similarity between the predicted and ground-truth masks. 

During testing, overlapping patches of dimensions 448 × 448 were extracted from the image, with patches overlapping by 50 pixels along both the height and width dimensions. The model takes the crops as inputs and outputs probability masks of dimensions 448 × 448 for each crop. Test Time Augmentation (TTA), which includes combinations of no-flip and horizontal flip along with rotations of 0°, 90°, 180°, and 270° are performed on each cropped patch. The application of the 8 augmentations gives us 8 augmented crops which were then passed to the model to generate 8 probability masks. No other spatial or color-based augmentations were performed during testing. The probability masks are passed through the inverse of the augmentation operations to de-augment them. The resulting 8 de-augmented masks are averaged together to obtain the probability mask on the input image patch. After doing this for all patches generated for an image, the final probability masks are patched back together with the appropriate weighting over overlapping regions to obtain the full probability mask having the same dimensions as the input image.

#### 2.4.2. Classification 

After evaluating the performance of our segmentation models using the Intersection-over-union (IoU) metric, we selected the best model and used it to identify irregular networks, irregular network mimics, and similar network-like structures in a dataset of 1000 images. A hold-out test set of 30% of the images is separated from this dataset for final testing. The remaining 70% of the data is split into 5 folds for 5-fold cross-validation of the deep learning classification models.

To train the deep learning classification models we used a constant initial learning rate of $\lambda_{c}=$ 0.00001, batch size of 8, weight decay of γ=1e−3, and input size of 448 × 448. To prevent over-fitting, early stopping was performed if the validation loss did not improve for 10 epochs. 

Similar to the data augmentation techniques described in our segmentation pipeline, we apply various augmentations to the training images for our classification task. These augmentations included spatial transformations such as random shifting, scaling, horizontal flips, and vertical flips. However, we used fewer color-based augmentations and did not apply elastic transformations. The augmented image was either cropped or resized to 448 × 448 with equal probability to make the model robust to scale variations and to enable whole lesion classification if desired. The Binary Cross-Entropy (BCE) loss was used as the loss function for training the model. All models used for segmentation and classification were trained using the Adam [44] optimizer. 

Next, taking inspiration from the data folding strategy for stacked generalization in [39], we used the predicted probabilities on the five validation folds of the classification dataset to train five sets of conventional classifier models. Specifically, we used the validation set of the deep learning classifier for a fold to generate another set of 80% training and 20% validation data. The cascade generalization framework (Section 2.2), where the prediction probabilities of the deep learning classifier on a fold are concatenated with the inputs to the classifier and fed into another classifier, was used to train each type of conventional model over a training and validation set. Training each conventional classifier on the five validation sets generated by the folding strategy mentioned above, results in five models per type of conventional model used. It is important to note that these conventional classifiers are out-of-the-box models, and no extensive hyperparameter tuning was performed. Once trained, the predictions from the five models for each conventional classifier were averaged to obtain the final output probabilities on that type of classifier. To obtain the final predictions, and analysis of classification results, both with and without cascade generalization, a constant threshold of 0.5 is applied to all output probabilities.

## 3. Results

### 3.1. DL Segmentation

Table 1 shows the best overall pixel-based scores per architecture for the irregular-net segmentation task. The complete results for each architecture with all the encoders and under different levels of output thresholding can be found in Table A1. The highest IoU was obtained using the U-Net++ model with an EfficientNet-b4 encoder. The IoU appears low, but the ground-truth irregular-network masks used for assessment are precise annotations of a feature that is very diverse. Further assessment of the predicted mask using overlays shows that the model sometimes detects globules and pseudopod structures, and an example of this can be seen in Figure 3, where no irregular network was present in the input lesion.

Figure 4 shows the overlay for an image where the model detects irregular networks on the periphery and a normal network in the lesion center, increasing the number of FP pixels in the mask. The IoU for this image, approximately 0.28, is the mean per image IoU for the whole test dataset (Table 1). Figure 5 shows a high IoU of 0.78, with an extra network detected by the model. 

### 3.2. Classification

For classification with our level-0 models, i.e., using conventional classifiers or DL models alone, we see that the DL models and their ensembles perform slightly better in terms of melanoma detection accuracy compared to the ensemble of conventional models when the probabilities are thresholded at 0.5. This is shown in Table 2 and can also be observed in the confusion matrices presented in Figure A2 and Figure A3 in Appendix A. We also observe that the ensembles of DL-only and conventional-only models show an improvement in accuracy over the individual classifiers of each type. After ensembling the DL models, we see an improvement of almost 4% in accuracy and recall over the best conventional-only model, the conventional classifier ensemble. This trend is also reflected in the ROC curves presented in Figure A1, where we have a high AUC of 0.917 for the DL ensemble compared to 0.881 for the conventional-only ensemble.

Once the cascade generalization framework is used with the probabilities from the DL models, we always see an improvement in the number of melanoma true positives over the classifications made based on the DL models alone. Figure A2 shows this improvement, where, compared with the level-0 ResNet50 model, the melanoma true positives jump by 25 with the level-1 conventional ensemble model and by 35 with the level-1 random forest model, resulting in a higher recall for melanoma. This increase in recall can be seen in Table 3 where the ResNet50 model with random forest gives the highest recall of 0.893. This trend can be seen for all models in the cascade generalization framework and can be observed by looking at the confusion matrices in Figure A4, Figure A6 and Figure A8. The corresponding ROC curves in Figure A5, Figure A7 and Figure A9 show that the AUC and AUC(FPR > 0.4) of the ensembled level-1 models are almost as close as those of the DL-only level-0 models and their ensemble. Despite this, the AUC, AUC(FPR > 0.4), and recall are all highest for the level-1 random forest, which uses the ResNet50 (level-0) output probabilities. Comparing the metrics without cascade generalization in Table 2 and with cascade generalization in Table 3 shows improvements to all metrics, including accuracy. 

Table 4 and Table 5 present an overview of the results for the classification pipelines presented. Table 4 shows the models and classification pipelines with the highest accuracy across the different pipelines presented in this work. We see a 2% increase in accuracy with the best cascade generalization pipeline over the best DL-only approach (ensemble) and an increase of around 7% over the ensemble of conventional classifiers. Table 5 shows the models with the highest recalls for the classification pipelines presented, with a higher recall implying fewer missed melanomas. With cascade generalization, we see an improvement of around 11% over the DL-only approach (Efficientnet-B1) and an improvement of 13% over the conventional classifier (Neural Net). Therefore, we always achieve an improvement in recall, f1-score, and accuracy over the naïve models (DL-only and conventional-only) when using the presented cascade generalization framework.

The permutation feature importance scores, as described in Section 2.3, are calculated, and the top 10 features with and without cascade generalization are presented in Table 6 and Table 7, respectively. Surprisingly, with cascade generalization, the DL level-0 probabilities were only the second most important feature. The standard deviation of an object’s (non-overlapping distinct contours or blobs) color inside the lesion along the L-channel in the LAB color space ranks as the most important feature both with and without cascade generalization. Of the top ten features across both Table 6 and Table 7, eight features are common across both methods. The variation of “lightness”, i.e., the L channel within the lesion, within and without objects (network-like structures), is a critical feature regardless of cascade generalization. The ratio of irregular network pixels inside the lesion to the lesion area and the ratio of irregular network pixels outside the lesion to the lesion area, termed densities in Table 6 and Table 7, were also used in this study. This shows that more interpretable and explainable features such as the ones presented here need to be re-examined for skin lesion diagnosis. Extended lists with the top twenty features based on feature importance can be found in Table A2 and Table A3. 

## 4. Hardware and Software 

All models were trained on an Intel(R) Xeon(R) Silver 4110 CPU (2.10 GHz) with 64 GBs of RAM. We used an NVIDIA Quadro P4000 GPU with an 8 GB RAM for training. We used the Segmentation models library version 0.3.1 [34] with Pytorch version 1.12.1 [35] to implement our segmentation models and to calculate our segmentation metrics. Our classification models are implemented in Pytorch and our conventional classifier models and all classification metrics are calculated using the Scikit-learn version 1.1.3 [45] package.

## 5. Discussion

The best diagnostic results for images of melanoma [4,5] have been obtained using DL applied to whole images. DL results are superior to conventional image processing diagnostic results and can exceed those of domain experts. This whole-image deep learning approach fails to incorporate domain knowledge—the knowledge specialists use to diagnose melanoma in the clinic—primarily critical dermoscopic features. These include network irregularities, dots, globules, structureless areas, granularity, and other visible clues. 

An irregular network is a large-scale feature in one of two ways. First, the network varies. In other words, it is different in one portion of the lesion than in another. Second, the network is apparent only in a “zoomed-out” image, for example, in the case of angulated lines. These lines are wide, often dozens of pixels in width, a feature that may not be apparent to the convolution kernel, which may be only 3–5 pixels wide. Large-scale features are the most likely to be missed by deep learning, due to their complex nature and the variation of the feature over the whole lesion.

The findings of this study show that DL may still be at a stage where domain knowledge supplied by conventional learning is needed for optimum results (Figure 6). The results presented here show that irregular network features can boost the recall of melanoma classification by as much as 10% and its accuracy by 2% over DL-only models. We have yet to reach fusion equilibrium, the point at which training cases supply all of the information needed for best diagnosis, and domain knowledge may be ignored.

While many machine learning and deep learning approaches have been applied to medical datasets, the use of the resulting research in clinics is still lagging. Therefore, the exploration of frameworks [46] for integrating these models to aid clinics is very important. One reason for this is the black-box nature of deep learning models. We believe our research will help in the creation of explainable deep learning pipelines and aid their acceptance within clinics. Another factor that would contribute to improving data collection and acceptance would be the development of better human-computer interfaces such as the one presented in [47] and [48]. 

## 6. Conclusions

In this study, we trained a deep learning model to identify irregular networks using 487 unique examples of lesions annotated with irregular network structures in melanoma. We compared different deep learning architectures, with all architectures using augmentation. The architecture with the highest IoU score for irregular network identification was the U-Net++ architecture.

Features of the identified irregular networks were analyzed using the random forest classifier, linear SVM, RBF SVM, decision trees, and neural networks. Fusing these features with the deep learning results for a set of 1000 images of melanomas and benign lesions showed an improvement in the area under the curve for melanoma identification. At a probability cutoff of 0.5, 35 more melanomas were found with a fusion of deep learning and the random forest classifier using the cascade generalization framework. We believe that better hyperparameter tuning of the conventional models and a wider array of hand-crafted features may improve the results further. The adoption of deep learning models in the clinic may be advanced by contextual hand-crafted feature identification, which will also improve diagnostic results, in agreement with published surveys [50].

Therefore, more studies on fusing dermoscopic feature analytic results with deep learning results are warranted.

## Figures and Tables

**Figure 1 cancers-15-01259-f001:**
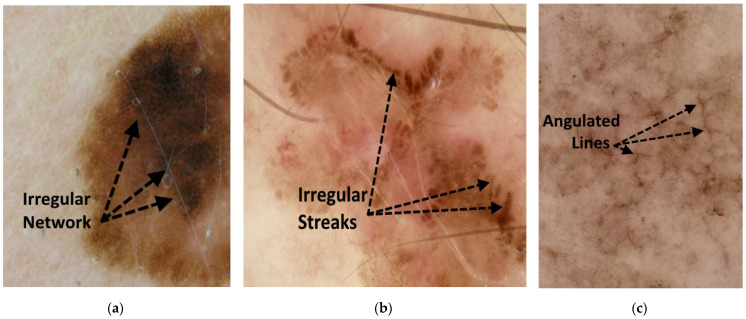
Figures showing examples of irregular network structures: (**a**) irregular network, (**b**) irregular streaks, and (**c**) angulated lines.

**Figure 2 cancers-15-01259-f002:**
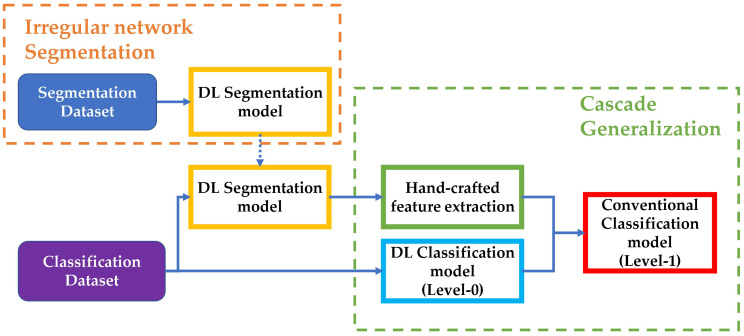
Overview of the steps involved in our implementation. The segmentation dataset is used to train and test the DL segmentation model. The segmentation model is used over the classification dataset to generate masks and extract hand-crafted features for level-1 of the Cascade generalization pipeline. The classification dataset is also used to train a DL classification model (level-0), which is used with the hand-crafted features to train conventional classifiers for the final diagnosis classification.

**Figure 3 cancers-15-01259-f003:**
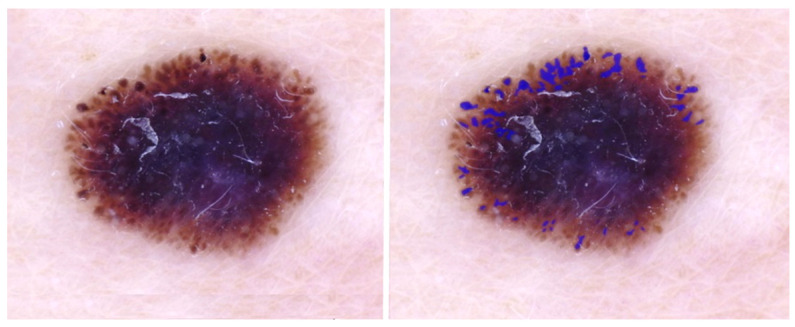
IoU = 0.0; DL detects extra structures: globules and pseudopods in the periphery. Blue regions signify false-positive areas.

**Figure 4 cancers-15-01259-f004:**
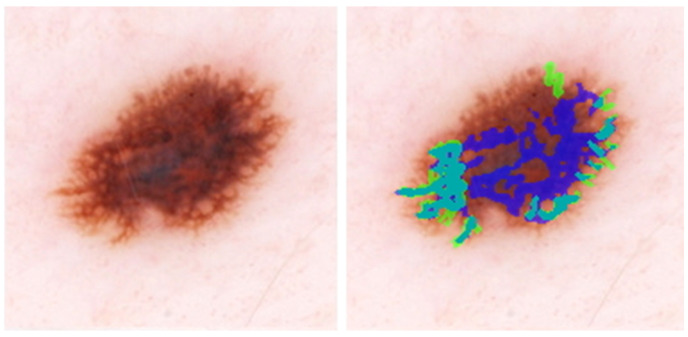
IoU = 0.29, close to the average IoU for all melanomas. DL detects peripheral irregular networks correctly. DL also detects normal network in the image center. Blue regions signify false-positive areas, green regions are false-negative, and teal-colored regions are true-positive.

**Figure 5 cancers-15-01259-f005:**
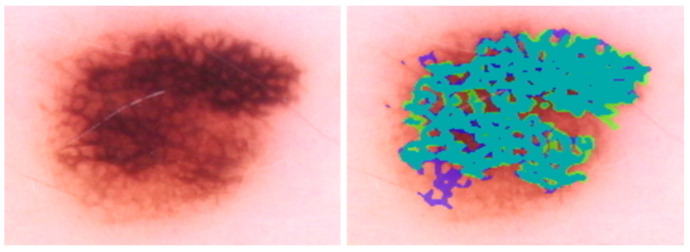
IoU = 0.78; DL can detect the areas of darkened lines and constricted holes in this melanoma. The primary DL error is at the lower left, where DL detects normal network. Blue regions signify false-positive areas, green regions are false-negative, and teal-colored regions are true-positive.

**Figure 6 cancers-15-01259-f006:**
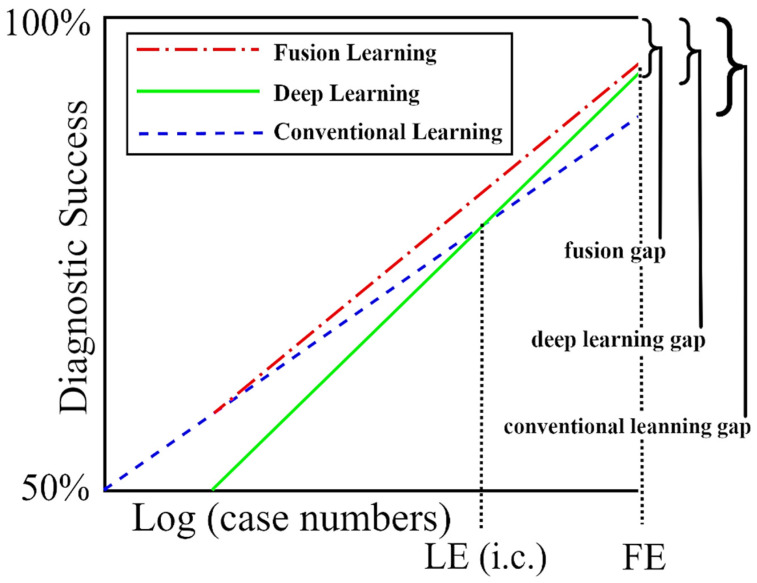
The success of deep learning (DL) with and without domain knowledge (conventional learning). As the number of training cases increases, DL becomes more accurate than conventional learning with LE (Learning Equilibrium) training cases, and finally becomes as accurate as fusion learning with FE (fusion equilibrium) training cases. At some future number of training cases, we reach a point where the deep learning gap equals the fusion gap. From [49], with permission.

**Table 1 cancers-15-01259-t001:** The table shows the pixel-based metrics ^1^ for the best encoder for each architecture based on the IoU score after applying a threshold of 0.5 on the model outputs.

Architecture	Encoder	Precision	Recall	F1-Score	Specificity	IoU
U-Net	EfficientNet-b4	0.363	0.523	0.428	0.984	0.273
U-Net++	EfficientNet-b4	0.358	**0.562**	**0.438**	0.982	**0.280**
MA-Net	EfficientNet-b5	**0.383**	0.461	0.419	**0.987**	0.265
PA-Net	EfficientNet-b5	0.345	0.501	0.408	0.983	0.257

^1^ Rounded to three significant digits. The highest value for each metric is in bold.

**Table 2 cancers-15-01259-t002:** The table presents the classification metrics for both deep learning (DL) models applied to lesion images and conventional models using only hand-crafted features. Additionally, the ensemble of DL architectures and conventional models, obtained by averaging the output probabilities, is presented. The metrics were calculated after applying a threshold of 0.5 on the output.

Level-0	Type	Precision	Recall	F1-Score	Accuracy
Efficientnet-B0	DL	0.842	0.787	0.813	0.817
Efficientnet-B1	DL	0.847	** 0.787 **	0.816	0.820
Resnet50	DL	** 0.899 **	0.686	0.779	0.802
Ensemble	DL	0.886	0.781	** 0.830 **	** 0.838 **
Decision Tree	Conventional	0.744	0.757	0.751	0.745
Linear SVM	Conventional	0.823	0.716	0.766	0.778
Neural Net	Conventional	0.806	**0.763**	**0.784**	0.787
Random Forest	Conventional	0.821	0.734	0.775	0.784
RBF SVM	Conventional	0.748	0.669	0.706	0.718
Ensemble	Conventional	**0.833**	0.740	**0.784**	**0.793**

The highest value for each metric for each type of model (DL or Conventional) is highlighted in bold, and the best across both types is underlined.

**Table 3 cancers-15-01259-t003:** The table presents the classification metrics for cascade generalization with DL models as level-0 models and conventional models as level-1 models. The metrics were calculated after applying a threshold of 0.5 to the probabilities.

Level-0	Level-1	Precision	Recall	F1-Score	Accuracy
Efficientnet-B1	Ensemble	0.832	**0.852**	0.842	0.838
Efficientnet-B1	Neural Net	0.811	0.811	0.811	0.808
Efficientnet-B1	Random Forest	**0.847**	**0.852**	**0.850**	**0.847**
Efficientnet-B1	Decision Tree	0.819	0.828	0.824	0.820
Efficientnet-B1	RBF SVM	0.796	0.692	0.741	0.754
EfficientNet-B1	Linear SVM	0.826	0.757	0.790	0.796
EfficientNet-B0	Ensemble	** 0.860 **	**0.870**	** 0.865 **	** 0.862 **
EfficientNet-B0	Neural Net	0.835	0.781	0.807	0.811
EfficientNet-B0	Random Forest	0.835	0.840	0.838	0.835
EfficientNet-B0	Decision Tree	0.830	0.811	0.820	0.820
EfficientNet-B0	RBF SVM	0.778	0.663	0.716	0.733
EfficientNet-B0	Linear SVM	0.841	0.751	0.794	0.802
Resnet50	Ensemble	0.849	0.834	0.842	0.841
Resnet50	Neural Net	0.830	0.781	0.805	0.808
Resnet50	Random Forest	0.825	** 0.893 **	**0.858**	**0.850**
Resnet50	Decision Tree	0.822	0.822	0.822	0.820
Resnet50	RBF SVM	0.772	0.663	0.713	0.730
Resnet50	Linear SVM	** 0.860 **	0.728	0.788	0.802

The highest value for each metric for each type of level-0 model is highlighted in bold, and the best across all cascade generalization pipelines is underlined.

**Table 4 cancers-15-01259-t004:** The table presents the classification metrics for the pipelines with the highest accuracy across each of the different pipelines used for classification. CG stands for cascade generalization.

Model	Classification Pipeline	Precision	Recall	F1-Score	Accuracy
Conventional Ensemble	No CG	0.833	0.740	0.784	0.793
DL Ensemble	No CG	**0.886**	0.781	0.830	0.838
EfficientNet-B0 + Conventional Ensemble	CG	0.860	**0.870**	**0.865**	**0.862**

The highest value for each metric is in bold.

**Table 5 cancers-15-01259-t005:** The table presents the classification metrics for the pipelines with the highest recall across each of the different pipelines used for classification. The highest value for each metric is highlighted in bold. CG stands for cascade generalization.

Model	Classification Pipeline	Precision	Recall	F1-Score	Accuracy
Neural Net	No CG	0.806	0.763	0.784	0.787
Efficientnet-B1	No CG	**0.847**	0.787	0.816	0.820
Resnet50 + Random Forest	CG	0.825	**0.893**	**0.858**	**0.850**

The highest value for each metric is in bold.

**Table 6 cancers-15-01259-t006:** The table shows the ten most important features based on the mean accuracy decrease after feature permutation. The scores are the averaged mean importance scores of all conventional classifiers without cascade generalization. Note that objects are non-overlapping distinct contours (blobs) in the irregular network binary mask generated by the segmentation model.

Feature	Importance Score
Standard deviation of object’s color in L-plane inside lesion	0.036
Mean of object’s color in L-plane inside lesion	0.026
Standard deviation of object’s color in B-plane inside lesion	0.026
Total number of objects inside lesion	0.021
Mean of skin color in A-plane (excluding both lesion and irregular networks)	0.019
Mean of object’s color in B-plane inside lesion	0.018
Standard deviation of skin color in L-plane (excluding both lesion and irregular networks)	0.016
Maximum width for all objects *	0.014
Standard deviation of skin color in B-plane (excluding both lesion and irregular networks)	0.012
Standard deviation of eccentricity for all objects	0.011

* Feature is not present in Table 7.

**Table 7 cancers-15-01259-t007:** Table shows the 10 most important features based on the mean accuracy decrease after feature permutation. The scores are the averaged mean importance scores for all conventional classifiers with Cascade Generalization. Note that objects are non-overlapping distinct contours (blobs) in the irregular network binary mask generated by the segmentation model.

Feature	Importance Score
Standard deviation of object’s color in L-plane inside lesion	0.037
Deep Learning (level-0) probability output *	0.030
Mean of skin color in A-plane (excluding both lesion and irregular networks)	0.030
Total number of objects inside lesion	0.021
Total number of objects remaining after applying erosion with circular structuring element of radius 3 *	0.015
Mean of object’s color in L-plane inside lesion	0.014
Standard deviation of skin color in B-plane (excluding both lesion and irregular networks)	0.014
Standard deviation of object’s color in B-plane inside lesion	0.014
Standard deviation of skin color in L-plane (excluding both lesion and irregular networks)	0.013
Mean of object’s color in B-plane inside lesion	0.012

* Features are not present in Table 6.

## Data Availability

Links to masks determined for these data may be found at https://zenodo.org/record/7557347.

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
