# Peer review of "Improving Automatic Melanoma Diagnosis Using Deep Learning-Based Segmentation of Irregular Networks"

_cancers, 2023, doi:10.3390/cancers15041259_

Round 1

Reviewer 1 Report

Overall, the paper presents an interesting and relevant topic in the field of deep learning for melanoma diagnosis. The approach of detecting irregularities in pigment patterns and combining it with unsupervised whole-image deep learning to improve diagnostic accuracy is novel and has the potential to make a significant contribution to the field. However, there are a few concerns that should be addressed before acceptance.

Firstly, the paper could benefit from more information about the context and scope of the work. The introduction presents a classifier that combines deep learning image-level results with 25 conventional image processing metrics for irregular pigment networks. However, for a better understanding of the research, the authors should provide more information on the specific gap in the field that their work aims to address. Additionally, it would be helpful if they clearly defined the research question and objectives of the study.

Secondly, the paper could benefit from more details on the theory and methodology used to develop the deep learning approach. A section describing the considered theory and applied methodology for developing the approach would help to situate the contribution by relating the findings to the research of others (doi.org/10.1109/JBHI.2022.3227125, doi.org/10.1016/j.artmed.2022.102285). Additionally, the paper would benefit from references to existing research that has used similar or related approaches.

Thirdly, data extraction should also be discussed in terms of privacy and ethics. Privacy and ethics are important considerations in any research involving personal information or medical data. This is especially true in the field of medical imaging, where sensitive personal information is captured and stored. In this case, the concern is that the paper should address how the data was obtained (doi.org/10.1146/annurev-bioeng-110220-012203, doi.org/10.1145/3399715.3399744, doi.org/10.1016/j.jacr.2017.12.028), and how the privacy and ethical considerations were addressed. This could include information on the informed consent process, data storage and handling, and any privacy-enhancing techniques used to protect the data (doi.org/10.1038/s42256-021-00337-8, doi.org/10.1016/S1470-2045(19)30149-4, doi.org/10.1145/3132272.3134111).

Fourthly, the paper could benefit from more detailed results and analysis. The paper should present the results in a clear and concise manner, providing sufficient details to allow the reader to understand and evaluate the findings (doi.org/10.1016/j.media.2022.102702, doi.org/10.1016/j.ijhcs.2022.102922, doi.org/10.3390/ijerph18105479). The paper should also provide a thorough analysis of the results, interpreting the findings in the context of the research question and objectives, and discussing the implications of the results (doi.org/10.3390/electronics11091294, doi.org/10.1016/j.ijhcs.2021.102607, doi.org/10.1109/TIPTEKNO47231.2019.8972045).

Finally, the paper would benefit from proofreading to correct any spelling errors or grammatical issues. The paper needs a brief proofreading as some English phrases seem to be off. Be aware of tenses also. It would be wise to proofread the paper and correct these, as well as other spelling errors.

In conclusion, the paper presents an interesting and relevant topic in the field of deep learning for melanoma diagnosis, but there are a few concerns that should be addressed before acceptance. Addressing these concerns would better contextualize and emphasize the contribution of the presented research, and improve the overall quality of the paper.

Author Response

Dear Reviewer 1,

We thank the reviewer for their careful reading of the study and thoughtful comments. We have made an effort to provide more literature survey and context for the paper as suggested by the reviewer in the submitted revision and revision document.

We hope the changes and edits made will be to your satisfaction.

Regards,

Anand K Nambisan

Reviewer 2 Report

The authors of the paper "Improving Automatic Melanoma Diagnosis using Deep Learning-based Segmentation of Irregular Networks" present several models for melanoma detection and classification using deep learning and machine learning methods such as U-Net.

The following are the comments on the article:

1. The proposed method must include a diagram showing the interconnection of the various DL-based blocks used in the study (Section 2.2).

2. The contribution of the study must be clearly stated (Abstract and Introduction sections). Re-write the Abstract.

3. Include the equations of the metrics used (Section 2.3).

4. Better organize the article to focus on the most important contributions and results. The paper is very long and it is not easy to find the naïve model versus the improved one.

5. There is bias in the models, deepen the discussion of these results. How convenient is it for a patient who is healthy to be classified as having melanoma?

6. To avoid confusion with the word "network" associated with melanoma vs. that associated with the deep learning model, it is suggested to use "architecture" for the second case.

7. Remove repeated sentences, e.g., "We use the predicted probabilities on the five validation folds of the classification dataset to train five sets of conventional classifier models" (lines 352-353 and 353-355).

Author Response

Dear Reviewer 2,

We are extremely thankful for the suggestions by the reviewer, we believe the revisions suggested have greatly improved the clarity and quality of the paper. We have added additional information and have re-organized the results section as suggested by the reviewer.

We hope the resulting revisions are to the reviewers satisfaction.

Regards,

Anand K Nambisan

Round 2

Reviewer 2 Report

The authors have updated the document according to the comments.  It is now ready for publication.